# Effective Disjoint Representational Learning for Anatomical Segmentation

**Priya Tomar**[*1,2]                                    PRIYA.PRIYA@IAIS.FRAUNHOFER.DE
**Aditya Parikh**[*1]                                    ADITYA.PARIKH@IAIS.FRAUNHOFER.DE
**Philipp Feodorovici**[3]                               PHILIPP.FEODOROVICI@UKBONN.DE
**Jan Arensmeyer**[3]                                    JAN.ARENSMEYER@UKBONN.DE
**Hanno Matthaei**[3]                                    HANNO.MATTHAEI@UKBONN.DE
**Christian Bauckhage**[1,2]                 CHRISTIAN.BAUCKHAGE@IAIS.FRAUNHOFER.DE
**Helen Schneider**[†1]                         HELEN.SCHNEIDER@IAIS.FRAUNHOFER.DE
**Rafet Sifa**[†1,2]                                RAFET.SIFA@IAIS.FRAUNHOFER.DE

[1] *Fraunhofer IAIS, Germany*

[2] *University of Bonn, Germany*

[3] *University Hospital Bonn, Germany*

**Editors:** Accepted for publication at MIDL 2025

## Abstract

In the wake of the limited availability of pertinent datasets, the application of computer vision methods for semantic segmentation of abdominal structures is mainly constrained to surgical instruments or organ-specific segmentation. Multi-organ segmentation has the potential to furnish supplementary assistance in multifarious domains in healthcare, for instance, robot-assisted laparoscopic surgery. However, in addition to the complexity involved in discriminating anatomical structures due to their visual attributes and operative conditions, the representation bias pertaining to organ size results in poor segmentation performance on organs with smaller pixel proportions. In this work, we focus on alleviating the influence of representation bias by involving different encoder-decoder frameworks for learning organ-specific features. In particular, we investigate the effect of organ-specific decoders on binary segmentation of anatomical structures in abdominal surgery. Additionally, we analyze the effect of organ-specific pretraining on the multi-label segmentation in two model training settings including knowledge sharing and disjoint learning, in relation to the contextual feature sharing between organ-specific decoders. Our results illustrate the significant gain in segmentation performance by incorporating organ-specific decoders, especially for less represented organs.

**Keywords:** Semantic segmentation, robot-assisted laparoscopic surgery, organ segmentation, Dresden surgical anatomy dataset.

## 1. Introduction

The applications of deep learning (DL) approaches have exhibited unforeseen advancements in medical image segmentation (Wang et al., 2022) (Qureshi et al., 2023)(Ye et al., 2023). Nonetheless, the limited availability of pertinent datasets due to the restrictions involved in medical data processing and requirement of domain-specific expertise in annotating the

---

[*] Contributed equally

[†] Contributed equally

intricate data features have restricted the developments to application-specific approaches. Medical imaging segmentation datasets predominantly focus on single-organ segmentation or specific anatomical regions (Rister et al., 2020), typically in computed tomography (CT) scans, magnetic resonance imaging (MRI) or X-Ray modalities (Fu et al., 2021)(Ji et al., 2022) (Schneider et al., 2023)(Häntze et al., 2024). In the domain of minimally invasive surgery, though datasets like EndoVis (Allan et al., 2020) have contributed to the significant developments on surgical instruments or organ-specific segmentation, datasets containing multi-organ annotations remain scarcely available (Carstens et al., 2023)(Hong et al., 2020) (Zhang et al., 2020). Furthermore, the surgical environment characteristic involving variable organ appearances, dynamic viewing conditions, frequent camera movements, and organ occlusions, present challenges in the extensive developments of DL-based methods (Rueckert et al., 2024).

For multi-organ segmentation in surgical data, current methods primarily focus on organ-specific learning. Kolbinger et al. (2023) investigated structure-specific models in comparison to common encoder and structure-specific decoder approach, reporting better performance for former. Similarly, Maack et al. (2024) focused on multi-teacher knowledge distillation (MT-KD) approach involving organ-specific decoders to learn through guided features of anatomy-specific teacher networks. Jenke et al. (2024) introduced an implicit learning method which emphasizes only the annotated classes in the images and considering missing ones as unknowns. It illustrated superior performance in comparison to organ-specific ensemble model but inferior when compared with fully-supervised learning for multi-organ segmentation. Notwithstanding the improvements in organ-specific performance, these approaches face challenges due to the class imbalance. For instance, the Dresden surgical anatomy (DSA) dataset (Carstens et al., 2023) contains organ pixel percentages ranging from 27.32% for abdominal wall to 1.25% for intestinal veins which often leads to learning bias towards the highly represented organs, thereby neglecting smaller or less frequent structures.

To improve the performance for underrepresented or intricate classes in laparoscopic images, Sinha and Dolz (2021) proposed a multi-scale attention mechanism combined with semi-supervised learning and effectively leveraged a small set of labeled laparoscopic images alongside unlabeled data. Similarly, Qiu et al. (2022) introduced a class-wise confidence-aware active learning framework which focused on dynamically selecting informative samples and exploiting unlabeled data, achieving significant improvements. (Zhang et al., 2024) propose a method that integrates class-wise contrastive learning with multi-scale feature extraction. By leveraging classification labels and employing a multi-scale projection head, the model effectively learns representations even with limited annotated data.

Our work builds on recent advancements in surgical image segmentation, specifically addressing key challenges in laparoscopic imaging and organ-specific learning for multi-organ segmentation. Drawing motivation from (Kolbinger et al., 2023), we focus on tackling the complexities arising from anatomical variability and inter-organ relationships in surgical imaging datasets. The significant variance in pixel proportions across organs in our dataset, where seven organs comprise less than 7% of the relative pixel area, necessitates a careful examination of organ-specific feature learning approaches. Our primary contributions include: (1) a comprehensive evaluation of four different segmentation architectures, including convolution neural network (CNN), transformers, and hybrid models for segmentation of

abdominal organs; (2) an in-depth analysis of three different training frameworks for investigating the influence of organ-specific feature learning in relation to model capacity; (3) investigation of effective intra-organ feature knowledge sharing between organ-specific decoders, incorporating both knowledge-sharing and disjoint learning frameworks for multi-label segmentation.

The application of our proposed methodology extends beyond surgical organ segmentation to other medical imaging domains. The organ-specific feature learning and knowledge sharing frameworks could benefit areas such as brain tumor segmentation (Chen et al., 2023), cardiac chamber analysis (Zhang et al., 2023), and musculoskeletal imaging (Wang et al., 2023) where anatomical structures exhibit similar challenges of size variation and complex spatial relationships.

## 2. Methodology

### 2.1. Dataset

We utilize the DSA dataset (Carstens et al., 2023), a high-resolution dataset specifically curated for computer-aided surgical applications and machine learning approaches in medical imaging. It comprises of 13,195 laparoscopic images from 32 real-world surgeries, with a minimum of 20 surgeries and 1000 images for each organ. It contains organ-specific binary segmentation masks for eleven anatomical structures including abdominal wall, colon, inferior mesenteric artery, intestinal veins, liver, pancreas, small intestine, spleen, stomach, ureter, and vesicular glands. Additionally, it contains a multi-class subset of 1430 images which are extracted from stomach subset and frequently display other organs, in total seven abdominal organs (abdominal wall, colon, liver, pancreas, small intestine, stomach and spleen). It exhibits significant class imbalance, with organ pixel percentages ranging from 27.32% for abdominal wall to 1.25% for intestinal veins in binary masks. Refer to (Carstens et al., 2023) for additional details concerning the dataset.

### 2.2. Training Framework

We investigate two architectural paradigms for multi-organ segmentation including Common Encoder-Common Decoder (CECD) and Common Encoder-Multiple Decoder (CEMD). This assessments aims to determine the effectiveness of organ-specific decoders in comparison to the standard multi-class segmentation approach for complex abdominal organ segmentation tasks.

- Common Encoder-Common Decoder (CECD): A single pipeline processes input images, producing multi-channel output corresponding to organ segmentation masks (Fig. 1a). This approach shares features across organs while maintaining efficiency. To distinguish between architectural and capacity effects, we include Expanded CECD (E-CECD) with $4\times$ wider decoder channels, following research that shows increasing channels width improve representation capacity without excessive computational cost(Zagoruyko and Komodakis, 2016)(Tan and Le, 2019).

- Common Encoder-Multiple Decoder (CEMD): This architecture uses a shared encoder with dedicated decoders for each of the eleven target organs (Fig.1a). The common

encoder captures general abdominal anatomy, while specialized decoders focus on organ-specific features and output organ-wise binary masks.

To assess the effectiveness of adopted frameworks, we consider five representative backbones including pure CNN: U-Net (Ronneberger et al., 2015) and DeepLabv3(Chen et al., 2017), transformer: SegFormer (Xie et al., 2021), and hybrid architectures: PVT U-Net (Zhu et al., 2023) and Attention U-Net (Oktay et al., 2018). U-Net and AU-Net use symmetric encoder-decoder structures, with AU-Net adding attention gates to focus on relevant spatial regions. PVT U-Net integrates vision transformer concepts with U-Net's structure, while SegFormer combines a hierarchical transformer encoder with an all-MLP decoder. DeepLabv3 applies atrous convolutions in parallel or cascade and capture multi-scale context with different atrous rates.

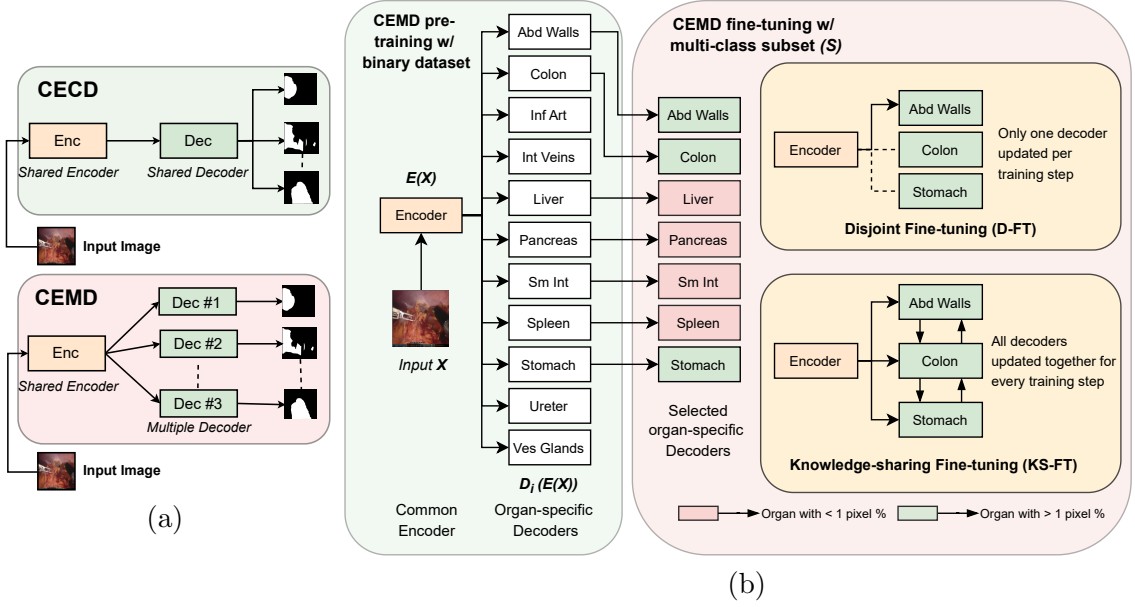

Figure 1: (a) Comparison of common-decoder (CECD) versus multiple-decoder (CEMD) frameworks. (b) Two fine-tuning strategies: Disjoint Fine-tuning (D-FT) where decoders update independently, and Knowledge-sharing Fine-tuning (KS-FT) with simultaneous decoder updates, both showing pre-training and fine-tuning phases for organ segmentation[†].

### 2.3. Parameter Sharing Strategies

We investigate two parameter sharing strategies for fine-tuning the CEMD framework for multi-class organ segmentation (Fig. 1b).

- Knowledge Sharing Fine-Tuning (KS-FT): It facilitates cumulative knowledge sharing by simultaneously updating all decoders during fine-tuning. It allows decoders to leverage inter-organ contextual knowledge, particularly beneficial for structures with similar visual attributes and spatial arrangements. The loss function for KS-FT is:

$$\mathcal{L}_{\text{KS}} = \sum_{i \in \mathcal{S}} \mathcal{L}_{\text{seg}}(D_i(E(X)), Y_i) \tag{1}$$

where $E(X)$ is the encoder output, $D_i$ is the $i$-th decoder, $Y_i$ is the ground-truth for organ $i$, and $\mathcal{S}$ represents set of all organs in the dataset. The segmentation loss $\mathcal{L}$seg is defined as the DICE loss function in this case.

- Disjoint Fine-Tuning (D-FT): It employs disjoint learning by updating each decoder while freezing others in a sequential manner. This preserves specialized anatomical features learned during initial training, maintaining organ-specific boundary characteristics. The loss function for D-FT is:

$$\mathcal{L}_{\text{Disjoint}} = \mathcal{L}_{\text{seg}}(D_k(E(X)), Y_k) \quad \text{for } k \in \mathcal{S}_X \tag{2}$$

where $\mathcal{S}_X$ represents set of organs present in the training sample $X$, corresponding to the multi-class subset in this case.

These contrasting approaches address a key trade-off between preserving specialized features and leveraging shared anatomical knowledge—particularly important for laparoscopic imaging where varying viewpoints and tissue deformation benefit from both specialized and shared feature learning.

### 2.4. Experiment Configuration

For data pre-processing, we resize the input to $256 \times 256$ and use standard data augmentation approaches during training including color jittering, random rotation, and image resizing to enhance model robustness and prevent overfitting. We use U-Net, PVT U-Net, and AU-Net with channel depths [64, 128, 256, 512], SegFormer with MiT-b0 backbone, and DeepLabv3 with Resnet50 backbone. We initialize all the models with random weights using fixed seed values to ensure reproducibility. For all the experiments, we use a batch size of 4 samples, Adam optimizer with an initial learning rate of $1 \times 10^{-3}$, a step-based learning rate scheduler with weight decay factor 0.1 after every 50 epochs, and early stopping based on validation DICE score with a patience of 10 epochs. For SegFormer, we used a lower initial learning rate of $1 \times 10^{-4}$ to enable stable gradient updates. We train models for 200 epochs using DICE loss (though convergence typically occurs earlier) and use it to select best model checkpoint. DICE loss exhibited superior performance in comparison to Binary Cross-Entropy (BCE) loss in our preliminary experiments (refer Appendix Table 5). Training is performed on an NVIDIA A100 GPU (40GB VRAM) using PyTorch framework.

### 2.5. Evaluation

For data splitting, we follow the recommended split by (Kolbinger et al., 2023) to ensure comparability in results. The training set (surgeries 1, 4–6, 8–10, 12, 15–17, 19, 22–25, 27–31), validation set (surgeries 3, 21, 26), and test set (surgeries 2, 7, 11, 13, 14, 18, 20, 32). In aggregate, the training validation and test sets containing binary organ masks subsume 7789, 1978 and 3328 samples respectively. For the multi-class subset, we focused on organs with foreground-to-background percentages exceeding 1% including the abdominal wall, colon, and stomach. The multiclass sets comprise 863, 202, and 365 samples in the training, validation and test sets respectively. Refer Table 4 in appendix for pixel ratios in different splits. As performance metrics, we report the DICE Score and Intersection over

Union (IoU) on held-out test set, as these metrics provide measure of spatial overlap while being robust to class imbalance.

## 3. Results and Discussion

### 3.1. Comparison of Backbone Models

For primary analysis, we deliberate over DICE scores and present the results of CEMD framework for our considered models in Figure 2. Refer to Table 6 and Table 7 in appendix for IoU and DICE scores. The results demonstrate that the AU-Net consistently outperforms other models for most anatomical structures, achieving superior performance in 8 out of 11 organs. The performance comparison between U-Net and AU-Net reveals that incorporating attention gates in the decoder pathway significantly improves segmentation accuracy. This improvement is particularly pronounced in organs with thin anatomical boundaries with distinctive features, such as the Ureter (+22.76%) and Vesicular Glands (+15.02%) Figure 5 in appendix illustrates the prediction masks for these two models.

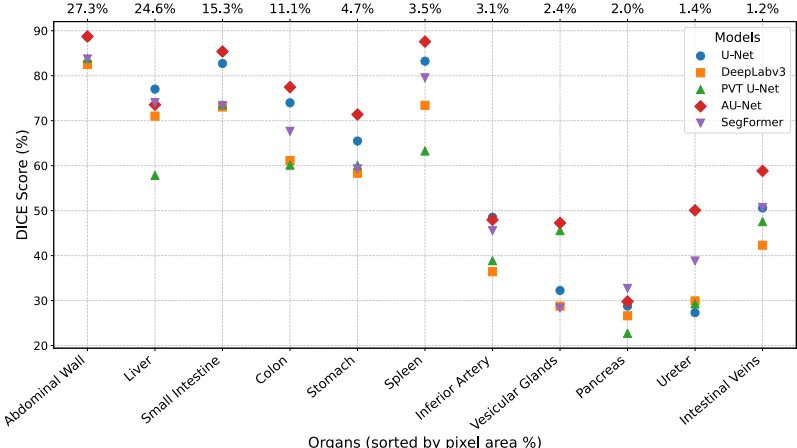

Figure 2: Mean DICE scores of models on different organs, sorted according to foreground to background pixel-ratio in the training data. AU-Net outperforms other models in majority of the cases. Mostly the influence of pixel area is observed in segmentation performance. Besides, the visual features of organ also influence performance, for instance, comparatively good performance of models on spleen in comparison to larger organs like liver and small intestine.

As anticipated, segmentation performance correlates with the organ pixel percentage (%) (foreground-to-background ratio). To investigate this, we computed Pearson (Kirch, 2008) and Spearman correlation coefficients (Spearman, 1904) between pixel % and DICE scores for each model. The results reveal a positive correlation between pixel % and DICE score for all models, with Pearson correlation coefficients of 0.64, 0.76, 0.48, 0.59, and 0.69 for U-Net, DeepLabv3, PVT U-Net , AU-Net and, SegFormer, respectively (Appendix Fig 4). These findings highlight the influence of class imbalance on model performance, underscoring the

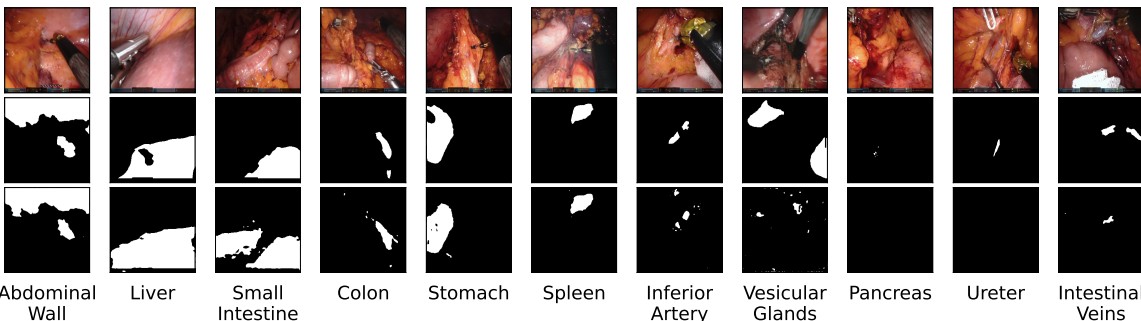

Figure 3: Segmentation results for AU-Net trained with common encoder multiple decoder (CEMD) architecture. Rows indicate input, target mask and prediction mask for 11 organs arranged in decreasing order of their respective pixel ratios in the training set. According to the evaluation metrics, model achieved best DICE and IoU scores on abdominal wall and lowest on Pancreas which is reflected in the prediction masks. Model is effective in detecting organs with distinctive visual attributes such as spleen, colon, inferior artery, and intestinal vein.

inherent challenge of segmenting smaller or less frequently visible structures in laparoscopic imagery.

Furthermore, it is evident that organ attributes are influential besides their pixel % in training data. The best performance of all the models is observed on abdominal wall, the organ with highest pixel-area. However, the scores of spleen (pixel % 3.5%) are higher than liver (pixel % 24.6%) for all 5 backbone models. Similarly, the least performance is observed for pancreas (pixel area 2%), not intestinal vein (pixel % 1.2%). Besides, the results delineates two categories of organs such that the models perform poorly on organs inferior artery, vesicular glands, pancreas, ureter, and intestinal veins in comparison to other organs with pixel % more than 3.5%.

### 3.2. Comparison of Base Architectures

Following our backbone model comparison, we focus on AU-Net due to its superior performance for comparing organ-specific decoder framework (CEMD) with common decoder frameworks (CECD and E-CECD) and present the DICE scores in Table 1. Refer to Table 8 in appendix for IoU scores. Table Table 1 reveal significant differences in both computational requirements and segmentation performance where the CEMD consistently outperformed CECD and its expanded variant E-CECD. While both E-CECD and CEMD have comparable parameter counts (152.83M and 156.83M respectively), their Giga Multiply-Accumulate Operations (GMACs) differ substantially (482.65 vs 443.71). This disparity in computational efficiency arises from how parameters are distributed across the network. In CEMD, the dedicated decoders process only their respective organ's features, whereas E-CECD's enlarged decoder processes all features through the same computational path, leading to increased multiply-accumulate operations. Per-image execution time measurements reinforce this efficiency distinction, with CECD (2.72 ± 0.15 ms) being significantly

faster than E-CECD (12.38 $\pm$ 0.11 ms) and CEMD (29.04 $\pm$ 0.37 ms), highlighting the computational cost of CEMD's superior segmentation performance. For additional analysis refer Appendix Table 8.

Table 1: Architectural Comparison: DICE Scores (in %) (Mean $\pm$ Standard Deviation) with pixel % and computational requirements for CECD, E-CECD, and CEMD variants. $+$ indicate an increase of at least $\approx 5\%$ in CEMD in comparison to E-CECD.

| Organ | Pixel (%) | CECD | E-CECD | CEMD |
|---|---|---|---|---|
| Abdominal Wall | 27.32 | 85.30 $\pm$6.79 | 86.39 $\pm$6.52 | **88.73** $\pm$5.73 |
| Liver | 24.59 | 70.09 $\pm$20.56 | 71.52 $\pm$17.64 | **73.55** $\pm$19.85 |
| Small Intestine | 15.32 | 82.48 $\pm$8.81 | 81.63 $\pm$7.42 | **85.38** $\pm$7.04 |
| Colon | 11.07 | 71.35 $\pm$14.15 | 71.06 $\pm$12.45 | **77.46** $\pm$10.26 $+$ |
| Stomach | 5.36 | 64.90 $\pm$17.35 | 66.93 $\pm$13.97 | **71.38** $\pm$16.57 $+$ |
| Spleen | 3.45 | 84.28 $\pm$7.89 | 79.06 $\pm$13.38 | **87.58** $\pm$6.98 $+$ |
| Inferior Artery | 3.14 | 36.42 $\pm$18.20 | 42.61 $\pm$15.68 | **47.95** $\pm$14.20 $+$ |
| Vesicular Glands | 2.37 | **47.41** $\pm$16.48 | 40.32 $\pm$19.52 | 47.27 $\pm$15.71 $+$ |
| Pancreas | 2.03 | **34.29** $\pm$19.97 | 27.47 $\pm$20.01 | 29.80 $\pm$18.68 |
| Ureter | 1.38 | 46.27 $\pm$17.81 | 43.17 $\pm$17.10 | **50.07** $\pm$19.24 $+$ |
| Intestinal Veins | 1.25 | 56.56 $\pm$19.65 | **63.89** $\pm$13.70 | 58.83 $\pm$15.64 |
| Overall | - | 61.78 $\pm$23.91 | 61.14 $\pm$23.93 | **65.27** $\pm$19.85 |
| Parameters (M) | - | 31.39 | 152.83 | 156.83 |
| GMACs | - | 55.95 | 482.65 | 443.71 |
| Memory Footprint (MB) | - | 893.95 | 2984.21 | 5442.54 |
| Execution Time (ms) | - | 2.72 $\pm$0.15 | 12.38 $\pm$0.11 | 29.04 $\pm$0.37 |

Despite similar parameter counts between E-CECD and CEMD, the latter demonstrates superior performance for 8 out of 11 organs. It reflects the significant influence of architectural design in achieving superior outcomes and effectiveness in capturing organ-specific anatomical features in comparison to the sheer number of model parameters. Notably, CEMD shows particular strength in segmenting organs with lower pixel percentages (particularly in inferior artery and ureter), suggesting its enhanced capability in handling class imbalance through its specialized architectural design. Concerning the organs, pancreas shows lowest scores which can be attributed to the challenges inherent in its detection due to limited field of view and indistinctive spatial boundaries.

### 3.3. Evaluating Parameter Sharing Strategies

We further analyze the influence of transfer learning and two different parameter sharing approaches including Knowledge Sharing Fine-tuning (KS-FT) and Disjoint Fine-tuning (D-FT) on multi-class set (introduced in Section 2.3). We consider AU-Net CEMD architecture as baseline and fine-tune it on a three-class organ subset subsuming abdominal wall, colon, and stomach.

Results (Table 2) demonstrate that D-FT consistently outperforms the baseline and KS-FT for all organs, achieving an overall improvement of 5.33% in DICE score after fine-tuning. It suggests that preserving organ-specific features during fine-tuning is crucial for maintaining segmentation accuracy. While KS-FT showed improvements for the abdominal wall and stomach in comparison to baseline, its performance degraded significantly for colon segmentation, indicating that simultaneous parameter updates may lead to feature interference in complex anatomical structures.

Table 2: Fine-tuning Strategy Comparison: DICE and IoU scores (in %) for base model, Knowledge Sharing Fine-tuning (KS-FT), and Disjoint Fine-tuning (D-FT) on three-organ subset.

|  |  | DICE Score (in %) | | | IoU Score (in %) | | |
|---|---|---|---|---|---|---|---|
| Organ | Pixel (%) | Base | KS-FT | D-FT | Base | KS-FT | D-FT |
| Abdominal Wall | 7.69 | 56.17 | 62.64 | **63.98** | 47.19 | 55.76 | **56.76** |
| Stomach | 6.97 | 65.21 | 65.52 | **66.22** | 55.26 | 55.80 | **56.59** |
| Colon | 5.60 | 33.38 | 21.42 | **38.56** | 25.76 | 15.16 | **29.55** |
| Overall | - | 51.59 | 49.86 | **56.92** | 42.07 | 42.24 | **47.63** |

## 4. Conclusion

Our work presents several contributions to the field of multi-organ segmentation in laparoscopic surgery. Firstly, our evaluation of backbone architectures demonstrates that AU-Net consistently outperforms conventional architectures. The integration of attention gates in standard U-Net proved particularly effective for organs with distinct boundaries. Secondly, our investigation of parameter sharing strategies revealed that architectural design, rather than model capacity alone, plays a crucial role in segmentation performance. The CEMD architecture, despite having similar parameter counts to E-CECD, achieved superior results, suggesting that organ-specific decoders are better suited for capturing unique anatomical features in surgical organ images. Our analysis of parameter sharing approaches for fine-tuning indicates the importance of preserving specialized features in transfer learning, particularly for complex anatomical structures.

Future work should address the persistent challenge of class imbalance between organ representations and segmentation performance. Additionally, investigating more sophisticated attention mechanisms and developing strategies for efficient knowledge transfer between organ-specific decoders could further improve performance for less-represented anatomical structures. Finally, exploring techniques such as Parameter-Efficient Fine-Tuning (PEFT) methods, including low-rank adaptations (LoRA), or pruning strategies, could be explored to optimize the architecture. These approaches have the potential to maintain high segmentation accuracy while significantly lowering computational requirements, making the model more practical for deployment in resource-constrained settings.

## Acknowledgments

This research has been funded by the Federal Ministry of Education and Research of Germany and the state of North-Rhine Westphalia as part of the Lamarr-Institute for Machine Learning and Artificial Intelligence, LAMARR22B. This research is funded by the Deutsche Forschungsgemeinschaft (DFG, German Research Foundation) under Germany's Excellence Strategy—EXC 2070-390732324-PhenoRob.

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

# Appendix

Table 3: Average Organ Ratios (foreground to background pixels) as percentages for Train, Test, and Validation Sets for binary segmentation dataset. The table highlights similar trends in pixel percentages across splits for each organ, demonstrating consistency in the data distribution. The rows are arranged according to the decreasing order of pixel ratios in the training set.

| Organ | Train Set (%) | Test Set (%) | Validation Set (%) |
|---|---|---|---|
| Abdominal Wall | 27.32 | 22.34 | 27.74 |
| Liver | 24.59 | 11.62 | 14.99 |
| Small Intestine | 15.32 | 15.99 | 16.26 |
| Colon | 11.07 | 14.04 | 11.75 |
| Stomach | 4.73 | 5.69 | 5.21 |
| Spleen | 3.45 | 2.19 | 4.00 |
| Inferior Artery | 3.14 | 2.19 | 2.51 |
| Vesicular Glands | 2.37 | 3.34 | 3.83 |
| Pancreas | 2.03 | 4.06 | 3.18 |
| Ureter | 1.38 | 0.85 | 1.18 |
| Intestinal Veins | 1.25 | 1.66 | 1.05 |

Table 4: Average Organ Ratios (foreground to background pixels) as percentages for Train, Test, and Validation Sets for multi-class segmentation subset arranged in the decreasing order of pixel ratio in training set. The table highlights inconsistencies in organ pixel distribution across splits, with only organs having pixel percentages greater than 1% in all splits (Abdominal Wall, Colon and Stomach) considered for evaluation. This ensures robustness, avoids bias from underrepresented organs, and addresses the issue of poor model training and performance on organs with extremely low pixel percentages.

| Organ | Train Set (%) | Test Set (%) | Validation Set (%) |
|---|---|---|---|
| **Abdominal Wall** | **13.99** | **7.67** | **19.29** |
| **Stomach** | **6.19** | **6.97** | **6.55** |
| **Colon** | **2.79** | **5.60** | **1.59** |
| Liver | 1.74 | 0.49 | 9.53 |
| Spleen | 0.60 | 0.00 | 0.00 |
| Small Intestine | 0.46 | 0.20 | 0.41 |
| Pancreas | 0.29 | 0.54 | 0.04 |

Table 5: Training Loss Function Comparison: DICE vs BCE, evaluating using DICE and IOU scores (in %) (Mean ± Standard Deviation) using the AU-Net architecture. The DICE loss function was chosen for its ability to better handle the imbalanced class distribution commonly present in medical image segmentation tasks. It emphasizes the overlap between predicted and ground truth masks, which is crucial when segmenting small, irregularly shaped organs where pixel-wise accuracy is less informative. As shown in the table, the use of DICE loss results in higher DICE and IoU scores for most organs compared to BCE loss, particularly for organs with smaller or more challenging shapes. The overall performance of DICE loss (65.27 ± 19.65 for DICE Score % and 52.78 ± 21.21 for IoU Score %) also outperforms BCE loss (57.63 ± 23.87 for DICE Score % and 45.35 ± 23.71 for IoU Score %), highlighting its superior sensitivity to challenging segmentation tasks. This is particularly important for precise delineation in medical imaging, where even small misclassifications can have significant clinical implications.

| Organ | DICE Score (%) | | IoU Score (%) | |
|---|---|---|---|---|
| | DICE Loss | BCE Loss | DICE Loss | BCE Loss |
| Abdominal Wall | **88.73** ±5.73 | 87.90 ±5.78 | **80.19** ±8.80 | 78.87 ±8.79 |
| Liver | 73.55 ±19.85 | **76.76** ±15.55 | 61.66 ±22.51 | **64.60** ±18.43 |
| Small Intestine | **85.38** ±7.04 | 82.41 ±7.11 | **75.11** ±10.25 | 70.67 ±9.79 |
| Colon | **77.46** ±10.26 | 72.56 ±12.81 | **64.27** ±12.82 | 58.40 ±14.65 |
| Spleen | **87.58** ±6.98 | 78.13 ±11.93 | **78.56** ±10.54 | 65.57 ±15.01 |
| Stomach | **71.38** ±16.57 | 63.82 ±16.79 | **57.92** ±19.01 | 48.97 ±17.29 |
| Inferior Artery | **47.95** ±14.20 | 40.75 ±13.75 | **32.64** ±11.85 | 26.51 ±10.74 |
| Intestinal Veins | **58.83** ±15.64 | 49.88 ±13.52 | **43.30** ±14.81 | 34.28 ±11.75 |
| Vesicular Glands | **47.27** ±15.71 | 26.86 ±14.32 | **32.34** ±13.60 | 16.33 ±9.98 |
| Pancreas | **29.80** ±18.68 | 25.77 ±17.50 | **19.10** ±14.60 | 16.06 ±12.71 |
| Ureter | **50.07** ±19.24 | 29.10 ±19.48 | **35.46** ±16.29 | 18.61 ±13.93 |
| Overall | **65.27** ±19.65 | 57.63 ±23.87 | **52.78** ±21.21 | 45.35 ±23.71 |

Table 6: Model Performance Comparison: DICE Scores (in %) (Mean ± Standard Deviation) across backbone architectures. Abbreviations: Abd Wall (Abdominal Wall), Inf Art (Inferior Artery), Int Veins (Intestinal Veins), Sm Int (Small Intestine), Ves Glands (Vesicular Glands). Foreground-to-background ratios: Abd Wall (27.32%), Liver (24.59%), Sm Int (15.32%), Colon (11.07%), Spleen (3.45%), Stomach (4.73%), Inf Art (3.14%), Int Veins (1.25%), Ves Glands (2.37%), Pancreas (2.03%), Ureter (1.38%).

| Organ | U-Net | DeepLabv3 | PVT U-Net | AU-Net | SegFormer |
|---|---|---|---|---|---|
| Abd Wall | 88.66 ±5.40 | 82.51 ±6.75 | 83.82 ±8.49 | **88.73** ±5.73 | 83.70 ±7.37 |
| Liver | **77.03** ±16.54 | 71.02 ±17.82 | 57.83 ±20.17 | 73.55 ±19.85 | 74.03 ±16.68 |
| Sm Int | 82.72 ±8.04 | 73.05 ±7.94 | 73.57 ±5.91 | **85.38** ±7.04 | 73.33 ±15.89 |
| Colon | 73.98 ±12.36 | 61.12 ±15.64 | 60.12 ±17.23 | **77.46** ±10.26 | 67.64 ±14.56 |
| Spleen | 83.23 ±9.11 | 73.42 ±10.82 | 63.26 ±19.67 | **87.58** ±6.98 | 79.52 ±13.08 |
| Stomach | 65.48 ±16.19 | 58.32 ±16.60 | 60.03 ±17.17 | **71.38** ±16.57 | 59.28 ±20.32 |
| Inf Art | **48.54** ±11.68 | 36.42 ±14.46 | 38.87 ±14.55 | 47.95 ±14.20 | 45.52 ±17.16 |
| Int Veins | 50.58 ±11.00 | 42.31 ±18.38 | 47.56 ±15.30 | **58.83** ±15.64 | 50.69 ±18.60 |
| Ves Glands | 32.25 ±15.14 | 28.76 ±13.54 | 45.56 ±15.39 | **47.27** ±15.71 | 28.34 ±16.67 |
| Pancreas | 28.77 ±18.07 | 26.62 ±14.93 | 22.72 ±18.99 | 29.80 ±18.68 | **32.71** ±24.10 |
| Ureter | 27.31 ±18.94 | 29.93 ±16.73 | 29.26 ±15.96 | **50.07** ±19.24 | 38.80 ±21.70 |
| Overall | 60.78 ±23.09 | 53.04 ±20.77 | 51.29 ±18.70 | **65.27** ±19.85 | 57.59 ±19.51 |

Table 7: Model Performance Comparison: IOU Scores (in %) (Mean ± Standard Deviation) across different backbone architectures.Abbreviations: Abd Wall (Abdominal Wall), Inf Art (Inferior Artery), Int Veins (Intestinal Veins), Sm Int (Small Intestine), Ves Glands (Vesicular Glands). Foreground-to-background ratios: Abd Wall (27.32%), Liver (24.59%), Sm Int (15.32%), Colon (11.07%), Spleen (3.45%), Stomach (4.73%), Inf Art (3.14%), Int Veins (1.25%), Ves Glands (2.37%), Pancreas (2.03%), Ureter (1.38%).

| Organ | U-Net | DeepLabv3 | PVT U-Net | AU-Net | SegFormer |
|---|---|---|---|---|---|
| Abd Wall | 80.02 ±8.21 | 70.73 ±8.95 | 72.96 ±11.30 | **80.19** ±8.80 | 72.63 ±10.47 |
| Liver | **65.14** ±18.69 | 57.83 ±17.82 | 42.56 ±11.77 | 61.66 ±22.51 | 61.19 ±18.70 |
| Sml Int | 71.29 ±11.18 | 58.14 ±9.58 | 58.53 ±7.38 | **75.11** ±10.25 | 60.15 ±18.23 |
| Colon | 60.14 ±14.63 | 45.70 ±15.22 | 45.07 ±17.17 | **64.27** ±12.82 | 52.87 ±16.10 |
| Spleen | 72.26 ±12.62 | 59.11 ±12.97 | 49.12 ±20.07 | **78.56** ±10.54 | 67.76 ±16.37 |
| Stomach | 50.77 ±17.53 | 42.94 ±15.35 | 44.97 ±17.16 | **57.92** ±19.01 | 44.95 ±19.77 |
| Inf Artery | 32.80 ±9.87 | 23.19 ±10.60 | 25.17 ±11.67 | **32.64** ±11.85 | 31.03 ±14.09 |
| Int Veins | 44.32 ±11.15 | 28.49 ±14.37 | 32.50 ±13.09 | **48.30** ±14.81 | 35.96 ±16.29 |
| Ves Glands | 20.26 ±11.54 | 17.55 ±9.56 | 30.80 ±13.18 | **32.34** ±13.60 | 17.69 ±12.23 |
| Pancreas | 18.25 ±13.86 | 16.25 ±10.52 | 14.20 ±12.96 | 19.10 ±14.60 | **22.21** ±18.61 |
| Ureter | 17.32 ±13.80 | 18.75 ±11.73 | 18.25 ±11.98 | **35.46** ±16.29 | 26.39 ±17.42 |
| Overall | 48.42 ±23.33 | 39.88 ±19.83 | 37.92 ±17.96 | **52.78** ±21.21 | 44.80 ±19.30 |

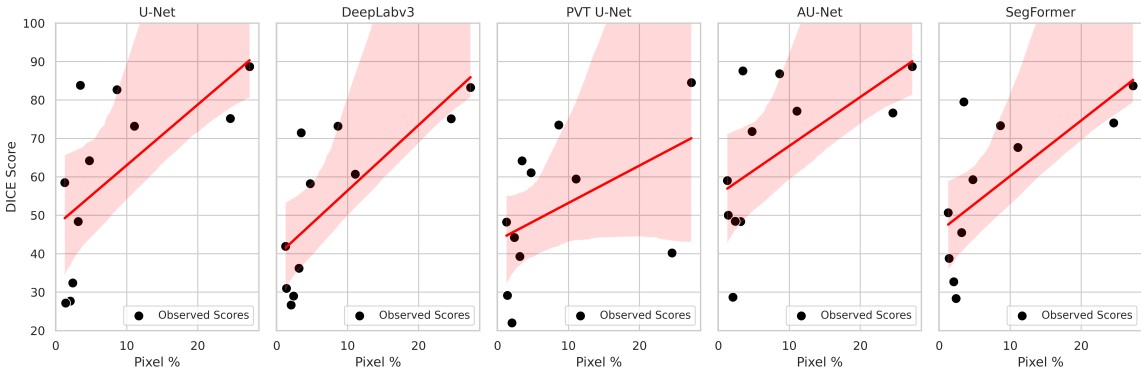

Figure 4: Scatter plots showing the correlation between pixel % and DICE scores (in %) for U-Net, DeepLabv3, PVT U-Net, AU-Net and SegFormer. Each plot includes a regression trendline and corresponding Pearson correlation coefficient.

Table 8: Architectural Comparison: IoU Scores (in %) (Mean ± Standard Deviation) and computational requirements for CECD, E-CECD, and CEMD variants.

| Organ | Pixel % | CECD | E-CECD | CEMD |
|---|---|---|---|---|
| Parameters (M) | - | 31.39 | 152.83 | 156.83 |
| GMACs | - | 55.95 | 482.65 | 443.71 |
| Abdominal Wall | 27.32 | 74.95 ±9.83 | 76.59 ±9.46 | **80.19** ±8.80 |
| Liver | 24.59 | 57.41 ±22.11 | 58.42 ±20.01 | **61.66** ±22.51 |
| Small Intestine | 15.32 | 71.07 ±12.02 | 69.62 ±10.42 | **75.11** ±10.25 |
| Colon | 11.07 | 57.22 ±16.04 | 56.45 ±13.98 | **64.27** ±12.82 |
| Stomach | 6.58 | 50.40 ±18.62 | 51.95 ±15.81 | **57.92** ±19.01 |
| Spleen | 3.45 | 73.62 ±11.60 | 67.21 ±16.86 | **78.56** ±10.54 |
| Inferior Artery | 3.14 | 23.84 ±14.25 | 28.36 ±12.97 | **32.64** ±11.85 |
| Vesicular Glands | 2.37 | **32.56** ±13.88 | 27.17 ±15.76 | 32.34 ±13.60 |
| Pancreas | 2.03 | **22.58** ±15.78 | 17.73 ±15.79 | 19.10 ±14.60 |
| Ureter | 1.38 | 31.87 ±15.44 | 29.00 ±13.61 | **35.46** ±16.29 |
| Intestinal Veins | 1.25 | 41.77 ±17.22 | **48.33** ±13.99 | 48.30 ±14.81 |
| Overall | - | 48.81 ±24.22 | 48.09 ±23.93 | **52.78** ±21.21 |

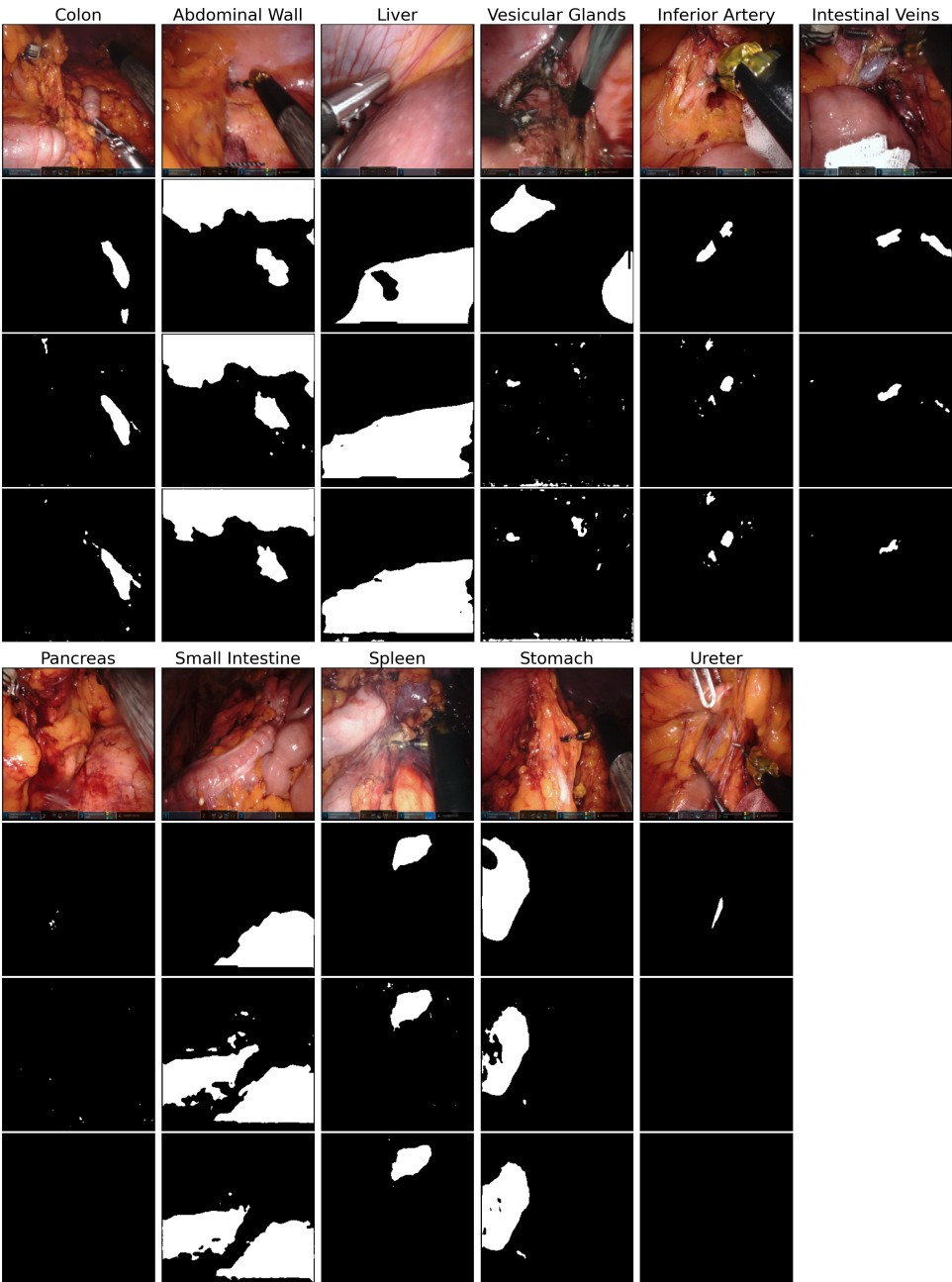

Figure 5: Qualitative comparison of segmentation results for 11 anatomical structures, comparing U-Net and AU-Net across two comprehensive visualizations. Each subplot includes (1) the original surgical image, (2) the ground truth segmentation mask, (3) U-Net predicted segmentation mask, and (4) AU-Net predicted segmentation mask. The first visualization presents results for the first six organs, while the second visualization shows results for the remaining five organs. AU-Net demonstrates superior boundary delineation and reduced false positives compared to U-Net, particularly in regions with complex anatomical interfaces.

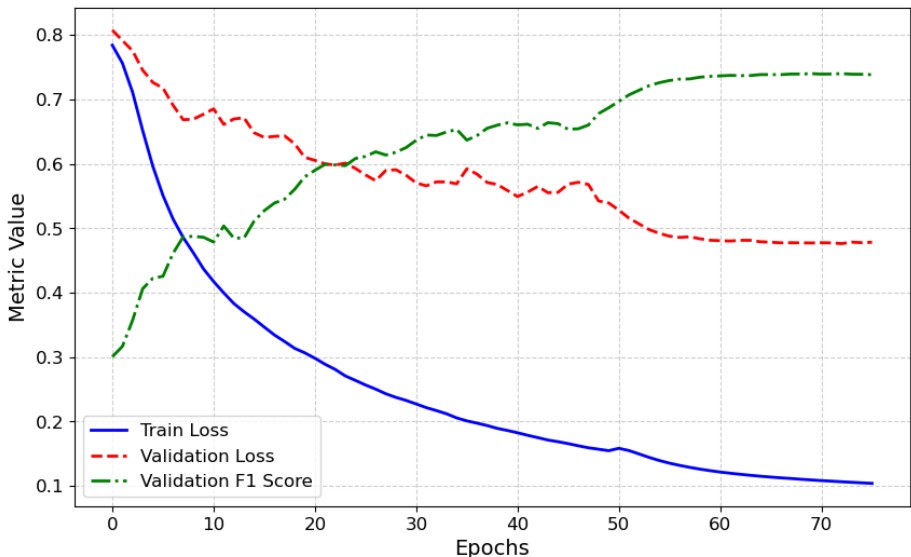

Figure 6: Segformer Training and Validation Loss with DICE (F1) Score. The plot demonstrates the smooth convergence of both training and validation loss, reflecting the effective optimization of the Segformer model. The steadily increasing validation DICE score indicates an improvement in model performance, while the closely aligned training and validation loss curves suggest minimal overfitting. This confirms the model's ability to generalize well over the epochs.

