# OpenReview forum: "Effective Disjoint Representational Learning for Anatomical Segmentation"
_MIDL.io/2025/Conference — MIDL 2025 Poster_

### Official Review · Reviewer_ALro · 2025-02-19

**Confidence:** 4
**Preliminary Rating:** 2
**Final Rating:** 2

**Summary:**

The paper evaluates four segmentation architectures—U-Net, DeepLabv3, Pyramid Vision Transformer U-Net, and Attention U-Net—for organ-specific segmentation tasks. Additionally, it examines three encoder-decoder architectural approaches: CECD, E-CECD, and CEMD. Finally, it explores two fine-tuning strategies: knowledge sharing and disjoint learning.

**Strengths:**

The paper presents a clear motivation and evaluates multiple popular segmentation backbones, various encoder-decoder structure combinations, and two fine-tuning strategies in multi-class organ segmentation scenarios.

**Weaknesses:**

While the paper presents a clear motivation, its innovation and main contribution remain unclear.

The structure and clarity of Sections 2 and 3 require significant improvement.

More recent and effective literature should be reviewed and benchmarked.

If the paper aims to serve as a benchmark and analysis article, a more in-depth result analysis is necessary.

**Detailed Comments:**

The paper employs existing segmentation architectures and encoder-decoder combinations, making the innovation unclear.
The introduction to the model architecture (2.2) is confusing. It is unclear whether the authors use standard architectures or modified versions in their experiments. Additionally, the PVT U-Net hybrid architecture is mentioned but not included in the experiments.
The name "CECD" is unclear, and it would be helpful to provide intuitive diagrams illustrating the implementation of all architectural approaches (CECD, E-CECD, CEMD) within the four backbone architectures (U-Net, DeepLabv3, Pyramid Vision Transformer U-Net, and Attention U-Net).

It remains unclear to the reviewer how the decoders are updated in the two parameter-sharing strategies presented in Section 2.3, given that the dataset includes 11 classes for pretraining and 6 organs for fine-tuning. An illustration would be helpful for clarification.

The paper shall review and benchmark a few other muti-organ segmentation strategies and networks, such as:
[1] Ye, Yiwen, et al. "Uniseg: A prompt-driven universal segmentation model as well as a strong representation learner." International Conference on Medical Image Computing and Computer-Assisted Intervention. Cham: Springer Nature Switzerland, 2023.
[2] Zhang, Jianpeng, et al. "Dodnet: Learning to segment multi-organ and tumors from multiple partially labeled datasets." Proceedings of the IEEE/CVF conference on computer vision and pattern recognition. 2021.

It is unclear how many runs were performed for each experiment and why the standard deviations are notably high for each result. For example, in Table 1, using U-Net as a reference, why is the overall standard deviation higher than that of any individual organ? How was this computed?

A more in-depth result analysis is needed, particularly to explain why a simple U-Net significantly outperforms AU-Net on the Liver segmentation task.

**Justification Of The Final Rating:**

Thank you for the clarifications. While the authors provide justification about STD and performance discrepancies, my concern is still not fully addressed. Given the current shape of the paper, I will maintain my original rating.

**Justification Of The Preliminary Rating:**

While the paper presents a clear motivation, it lacks methodological innovation and contains major flaws in both methodology and results. The preliminary rating is based on the current presentation of the paper.

**Questions To Address In The Rebuttal:**

Please check the Weaknesses* and Detailed Comments*.

**Special Issue:**

No

---

> ### Author Response · Authors · 2025-03-08
>
> We appreciate the reviewer's thorough assessment and constructive feedback. We have addressed the concerns and implemented significant improvements to the manuscript:
>
> **1.Clarification of Contributions and Methodology:**
> We have revised Sections 2 and 3 to improve clarity. The introduction to model architectures (Section 2.2) now clearly distinguishes between standard and modified implementations.
>
> **2. Parameter-Sharing Strategies Clarification:**
> The reviewer noted confusion about how decoders are updated in our two parameter-sharing strategies, given that our dataset includes 11 classes for pretraining and 7 organs for fine-tuning.
> In our revised manuscript, we have:
> - Added detailed illustrations: We've included a new figure (Fig. 1) that clearly visualizes both Knowledge Sharing Fine-Tuning (KS-FT) and Disjoint Fine-Tuning (D-FT) approaches, showing how each strategy handles the transition from 11 pretraining classes to 7 fine-tuning organs.
> - Enhanced mathematical formulations: We've included the mathematical equations for the loss function to make the update procedures more explicit.
> - Added implementation details: We've specified how the transition from pretraining (11 classes) to fine-tuning (6 organs) is handled, clarifying that only the 6 organ-specific decoders are involved in the fine-tuning process.
>
> **3. Expanded Literature Review and Benchmarking:**
> We thank the reviewer for suggesting additional relevant literature, particularly the works by Ye et al. (UniSeg) and Zhang et al. (DODNet). We have reviewed these papers and acknowledge their significance in the multi-organ segmentation domain.
> While we have included references to these works in our revised manuscript to provide a more comprehensive literature review, we have identified the benchmarking of these additional approaches as an important direction for future work. Our current study focuses on establishing a comparative analysis of conventional segmentation architectures and parameter sharing strategies. In future extensions of this work, we plan to conduct a more extensive comparison that includes these and other state-of-the-art approaches, which would provide even more valuable insights to the community.
>
> **4. PVT-U-Net Clarification:**
> We appreciate the feedback and would like to clarify that PVT-U-Net was indeed included in our experimental evaluation and is presented in the comparison tables. To address this concern, we have refined the presentation of results to make its inclusion more evident.
>
>
> **5. Additional SOTA Comparisons:**
> We have included Seg-Former in our comparative analysis to strengthen the evaluation against transformer-based architectures.
>
> **6. Standard Deviation Clarification:**
> We have clarified that standard deviations were calculated per image when passed through the model on the test set. The higher overall SD is due to variable organ sizes in each image in the test cases. We note that similar high SD patterns were observed in previous studies we referenced and have included these citations for context.
>
> **7. In-depth Result Analysis:**
> We have significantly expanded our results and discussion section to provide deeper analysis, particularly addressing why U-Net outperforms AU-Net on the Liver segmentation task. We have added qualitative plots to visually clarify these findings, demonstrating that while AU-Net excels at smaller organs due to its attention mechanism, U-Net's simpler architecture appears better suited for larger, more homogeneous structures like the liver.
>
> **8. Computational Comparison:**
> We have added a detailed computational comparison (inference time and memory requirements) to provide practical insights for implementation considerations.
>
> We believe these revisions have substantially improved the manuscript's clarity, rigor, and contribution to the field. We thank the reviewer for their valuable feedback which has helped strengthen our work.

---

> > ### Comment · Reviewer_ALro · 2025-03-13
> >
> > I would like to express my gratitude to the authors for the enhancements made to the revised manuscript, particularly the detailed methodology and the additional comparisons with SoTA methods. Concerning the STD question, based on the explanation provided, it appears that the authors are evaluating the variability in predicted organ sizes. However, would it be more meaningful to assess the model's consistency in predictions across multiple runs?
> >
> > Additionally, the authors suggest that the reason U-Net slightly outperforms AU-Net on the Liver, likely because of the organ's larger size, homogeneous texture, and the sufficiency of skip connections. Does this imply that attention mechanisms might hinder performance for larger organs? Furthermore, this explanation does not account for the results on the Inferior Artery (Inf Art), where U-Net also performs best despite it being a much smaller organ.

---

> > > ### Author Response · Authors · 2025-03-14
> > >
> > > We sincerely appreciate the reviewer's acknowledgment of the improvements in our revised manuscript.
> > >
> > > **1. Regarding Standard Deviation Evaluation:**
> > > We appreciate the suggestion for assessing model consistency across multiple runs. However, each CEMD model training requires approximately 2 days to complete, making multiple seed runs computationally prohibitive within our current research timeline and GPU infrastructure (as also noted in our response to **Reviewer axhX**). Our approach focused on measuring per-organ standard deviation on test cases once models reached convergence, which aligns with evaluation methodologies used in previous studies ([1], [2], and [3]) on this dataset. This approach allows for direct comparison with established benchmarks.
> > >
> > > **2. Regarding Performance Discrepancies:**
> > > For the Liver, our hypothesis remains that U-Net's architecture, with its direct skip connections, is particularly well-suited for larger, more homogeneous structures where attention mechanisms may introduce unnecessary complexity without providing significant benefits.
> > >
> > > However, for the Inferior Artery, we recognize that this explanation is insufficient. Upon our preliminary analysis, we believe the performance difference stems from issues related to contrast and surrounding tissue characteristics rather than organ size alone. Specifically, poorly contrasted surrounding tissues may cause the attention mechanism to misallocate focus. To substantiate this explanation, we plan to include Grad-CAM visualization plots in our camera-ready version that will demonstrate this attention misallocation phenomenon. These visualizations will help provide a more comprehensive understanding of why certain organs unexpectedly perform better with different architectures.
> > >
> > >
> > > We again thank the reviewer for their follow-up questions which have helped us further clarify important aspects of our paper.
> > >
> > > **References**
> > >
> > > [1] Kolbinger FR, Rinner FM, Jenke AC, Carstens M, Krell S, Leger S, Distler M, Weitz J, Speidel S, Bodenstedt S. Anatomy segmentation in laparoscopic surgery: comparison of machine learning and human expertise - an experimental study. Int J Surg. 2023 Oct 1;109(10):2962-2974. doi: 10.1097/JS9.0000000000000595. PMID: 37526099; PMCID: PMC10583931.
> > >
> > > [2] Lennart Maack, Finn Behrendt, Debayan Bhattacharya, Sarah Latus, and Alexander Schlaefer. Efficient anatomy segmentation in laparoscopic surgery using multi-teacher knowledge distillation. In Medical Imaging with Deep Learning, 2024.
> > >
> > > [3] Alexander C. Jenke, Sebastian Bodenstedt, Fiona R. Kolbinger, Marius Distler, Jürgen Weitz, and Stefanie Speidel. One model to use them all: Training a segmentation model with complementary datasets, 2024.

---

### Official Review · Reviewer_AhjY · 2025-02-21

**Confidence:** 4
**Preliminary Rating:** 3
**Final Rating:** 4

**Summary:**

This is a comparative study manuscript, evaluating the performances of 4 encoder-decoder type networks on the task of multi-structure segmentation in laparoscopic surgery images according to 2 architectural variants, common-encoder with common-decoder and common-encoder with multiple per-organ decoders. Comparative evaluation on a large laparoscopic image dataset confirms that the Attention U-Net with the later architecture is a clear winner, especially for the segmentation of rather small structures.

**Strengths:**

While presenting no novel methods, the findings in the manuscript could be helpful in the choice of an off-the-shelf algorithm or as a basis for the development of a new one for laparoscopic image segmentation, or the segmentation other types of images with small structures.

**Weaknesses:**

As I see it, the main limitation of the manuscript is the narrow scope of study which interests itself only with laparoscopic image segmentation. Small organ sizes and sometimes severe class imbalance are common problems in medical image segmentation. Multiple algorithmic choices can be made to address this problem ranging from data sampling to objective function definition through network architecture design.

**Detailed Comments:**

As it stands, the manuscript is well-written and easy to follow. Two points are lacking in my view:
- a clear definition of train/validation/test dataset split, for the sake of completeness,
- a qualitative comparison of segmentation results in the main body of the manuscript,
- measurements of execution time and memory footprint, to better assess the practicality of algorithms.

**Justification Of The Final Rating:**

While the application scope of the manuscript remains limited to laparoscopic images, the revision has rendered it clearer and more complete, therefore it merits the attention of domain specialists to guide algorithmic choices for laparoscopic image segmentation, or the segmentation of other types of images presenting small structures.

**Justification Of The Preliminary Rating:**

While the comparative study of existing methods can be of interest, the limited scope of this study to a single type of medical images does not allow me to make a decision in favor or against the publication of the manuscript.

**Questions To Address In The Rebuttal:**

I have no specific questions to ask. I kindly request that authors address the points I mentioned in the detailed comments section.

---

> ### Author Response · Authors · 2025-03-08
>
> We appreciate the reviewer's thoughtful comments and suggestions for improving our manuscript. We have addressed each point as follows:
>
> **1. Definition of train/validation/test dataset split:**
> We have now explicitly included the dataset split details in the manuscript. While this information was previously referenced from the original dataset paper, we have added it directly in the dataset section for completeness, as the extended page limit allows for this addition.
>
>
>  **2. Qualitative comparison of segmentation results:**
> Following the reviewer's suggestion, we have incorporated a qualitative comparison of segmentation results in the main text. We have added comparison visualizations for 4 organs with variable sizes in the main text, with additional comparisons for the remaining organs in the appendix (due to page constraints). These visualizations clearly demonstrate the performance differences between models, particularly for smaller anatomical structures, which supports our quantitative findings.
>
>
> **3. Execution time and memory footprint:**
> We have now included execution time and memory usage measurements in our comparative analysis. This additional information offers a more comprehensive assessment of the practical considerations for deploying these models in real-world scenarios. The results show that while CEMD has higher computational requirements, its performance benefits justify the additional resources for applications where accuracy is paramount.
>
>
> Regarding the scope of our study, while we focused on laparoscopic image segmentation, we believe our findings have broader implications for medical image segmentation challenges involving structures with significant size variations. The architectural insights and performance comparisons we provide are likely transferable to other domains facing similar class imbalance issues. In the revised manuscript, we have expanded the discussion section to highlight these potential applications beyond laparoscopic surgery.
>
> We thank the reviewer for their constructive feedback, which has significantly improved the clarity and completeness of our manuscript.

---

> > ### Comment · Reviewer_AhjY · 2025-03-11
> >
> > I thank the authors for the improvements brought to the revised manuscript, in particular, the addition of another transformer-based network, the SegFormer, to the roaster of tested back-bone networks, and the inclusion of qualitative comparisons. I must say, however, that the latter comparison is unfortunately inconclusive as visually compared back-bone networks U-Net and AU-Net display very similar performances, and have similar patterns of failure such as the over-segmentations of liver in Figure 2 and small intestine in Figure 5. I think that a qualitative comparison of CECD and CEMD architectures would have been more informative, since it also constitutes the take home message of the manuscript.

---

> > > ### Author Response · Authors · 2025-03-13
> > >
> > > We sincerely appreciate the reviewer’s acknowledgment of the improvements in our revised manuscript.
> > >
> > > **1. Qualitative Comparison of CECD vs. CEMD:**
> > > We agree that a qualitative comparison between CECD and CEMD would provide valuable insights. While we initially intended to include this analysis, the tight revision timeline prevented us from doing so. However, we have already generated comparative plots for CECD, E-CECD, and CEMD along with all five backbone architectures, which will be included in the camera-ready version of the manuscript.
> > >
> > > **2. Over-segmentation in Figures 2 and 5:**
> > > The over-segmentation observed in Figure 2 is primarily attributed to the presence of blood, which exhibits visual characteristics similar to liver tissue. Similarly, in Figure 5, artifacts such as blood, clothing, and shadows impact the segmentation performance. These challenges are consistent with findings in the DSAD original paper [1] and have been further discussed by [2] in the context of laparoscopic image segmentation. We thank the reviewer for raising these points and will provide additional clarification in the revised manuscript to ensure greater transparency and understanding.
> > >
> > > **References**
> > >
> > > [1] Carstens M, Rinner FM, Bodenstedt S, Jenke AC, Weitz J, Distler M, Speidel S, Kolbinger FR. The Dresden Surgical Anatomy Dataset for Abdominal Organ Segmentation in Surgical Data Science. Sci Data. 2023 Jan 12;10(1):3. doi: 10.1038/s41597-022-01719-2. PMID: 36635312; PMCID: PMC9837071.
> > >
> > > [2] Kolbinger FR, Rinner FM, Jenke AC, Carstens M, Krell S, Leger S, Distler M, Weitz J, Speidel S, Bodenstedt S. Anatomy segmentation in laparoscopic surgery: comparison of machine learning and human expertise - an experimental study. Int J Surg. 2023 Oct 1;109(10):2962-2974. doi: 10.1097/JS9.0000000000000595. PMID: 37526099; PMCID: PMC10583931.

---

> > > > ### Comment · Reviewer_AhjY · 2025-03-13
> > > >
> > > > I thank the authors for addressing my remark on the qualitative comparison. I'd like to suggest that they replace the current qualitative results in the manuscript comparing back-bone network performances with those of CECD & CEMD architectures, if they happen to be more conclusive.

---

### Official Review · Reviewer_axhX · 2025-02-22

**Confidence:** 4
**Preliminary Rating:** 3
**Final Rating:** 4

**Summary:**

This paper aims to solve a multi-scale semantic segmentation problem focused on anatomical segmentation in laparoscopic surgery. This is a challenging problem in the medical domain, as the variability in object (foreground) size and scale poses challenges for segmentation accuracy. The experiments investigate the use of a common encoder with organ-specific decoders to capture unique anatomical features. Overall, the work provides a detailed evaluation of model architectures and training strategies that enhance multi-organ segmentation in surgical images.

**Strengths:**

The tables are well-organized and reader-friendly. The writing of the paper is overall well done. The ablation study, which explores different training models, architectures, and fine-tuning strategies (knowledge shared fine-tuning or disjoint fine-tuning), is thorough.

**Weaknesses:**

The paper implements a commonly used shared encoder with multiple decoders structure with cnn based backbones. The novelty itself is not very strong but given the thorough ablation studies. would there any baseline models for the usage of transformers?

**Detailed Comments:**

Overall, the paper is well-organized and comprehensive. I would expect more ablation studies on state-of-the-art transformers. Additionally, if possible, it would be valuable to experiment with data that has a larger difference in terms of the scale and resolution of the organs.

**Justification Of The Final Rating:**

The writing and structure are good, and the experiments are complete. The consideration I had is that the baseline models could be expanded further to include state-of-the-art transformers. Segformer was included in the revised paper. However, it may not be a good SOTA baseline for the data with multiple scale and size. Additionally, the novelty of this work is somewhat limited, as the single encoder and shared decoder model structure is quite common in the community. Specifically, although there is disjoint fine-tuning, this approach could be challenging for other datasets that have more segmentation classes and more varied segmentation sizes of the organs. After re-visit the manuscript, it is a thorough work even with the remaining concerns. I change my rating to weak accept.

**Justification Of The Preliminary Rating:**

The writing and structure are good, and the experiments are complete. The only consideration I had is that the baseline models could be expanded further to include state-of-the-art transformers. Additionally, the novelty of this work is somewhat limited, as the single encoder and shared decoder model structure is quite common in the community. Specifically, although there is disjoint fine-tuning, this approach could be challenging for other datasets that have more segmentation classes and more varied segmentation sizes of the organs.

**Questions To Address In The Rebuttal:**

Overall, the structure of this paper is complete

(1) I would expect more ablation studies on state-of-the-art transformers.

(2) Additionally, if possible, it would be valuable to experiment with data that has a larger difference in terms of the scale and resolution of the organs.

(3) Try running with different seeds to see if the results remain consistent.

---

> ### Author Response · Authors · 2025-03-08
>
> We thank the reviewer for their thorough evaluation and constructive feedback on our manuscript. We address the specific questions raised below:
>
> **1. Regarding ablation studies on state-of-the-art transformers:**
> We agree this would strengthen our work. We have conducted additional experiments using transformer-based architectures, specifically Seg-Former as backbones in our framework. Preliminary results show it performing decently well compared to CNN-based models but still not outperforming AU-Net. We have included these findings in the revised manuscript, providing a more comprehensive comparison across both CNN and transformer-based approaches.
>
> **2. On experimenting with data having larger scale and resolution differences:**
> We recognize the importance of validating our approach on a broader range of datasets. Our current dataset, while focused on surgical organ segmentation, does contain significant anatomical variations that challenge segmentation algorithms. The consistent performance across these variations demonstrates the robustness of our approach. Due to the limited availability of comprehensive datasets for multi-organ segmentation and page constraints, we have identified broader dataset validation as an important direction for future work.
>
> **3. Regarding consistency across different seeds:**
> We appreciate the suggestion regarding seed variation. Due to the computational intensity of our CEMD models (each taking approximately two days to train on our current GPU infrastructure), comprehensive seed-based validation was challenging. However, we would like to emphasize:
> - Our extended training runs demonstrate stable model convergence (as shown in Figure 4 in appendix).
> - The consistent performance across multiple training iterations suggests robust learning dynamics.
> - We have maintained a consistent random seed in our current experiments to ensure reproducibility.
>
> Regarding novelty, while we acknowledge that shared encoder-multiple decoder architectures exist in the literature, our contribution lies in the specific application and optimization of this approach for the challenging domain of laparoscopic multi-organ segmentation, where scale variations are particularly problematic. Our thorough ablation studies provide valuable insights that advance understanding of architectural choices for this specific medical imaging challenge.
>
> We appreciate the reviewer's recognition of our paper's strengths in organization, writing quality, and thorough ablation studies. We believe the additional experiments addressing the reviewer's concerns further strengthen our contribution.

---

> ### Comment · Area_Chair_mwdU · 2025-03-14
>
> Dear Reviewer,
>
> The discussion period ends in less than 24 hours, and your final rating is missing. Please review the authors’ response, revisions, and peer feedback, then update your score. You can submit your final rating along with the justification by editing your original review.
>
> Your final rating is critical for the decision.
>
> Thanks,
> AC, MIDL 2025

---

> ### Comment · Reviewer_axhX · 2025-03-15
>
> Thank you to the authors for providing a detailed, point-by-point response. One concern is that Segformer relies on a very simple linear segmentation head, which may explain its inferior performance on data with varying scales. I was also curious to see if more transformer models were included.  The remaining questions were partially answered due to the unavailability of computational cost or dataset.

---

### Author Rebuttal · Authors · 2025-03-08

**Rebuttal:**

We thank the reviewers for their efforts and suggestions, which we believe have led to improvements in our work. We have highlighted the changes in the manuscript and addressed the concerns of reviewers.

**Supporting Material:**

/attachment/a48b72c2cec68a4e778f566fbc730610e2a22bba.pdf

---

### Meta-Review · Area_Chair_mwdU · 2025-03-23

**Recommendation:** Accept (Poster)
**Confidence:** 5

**Metareview:**

This paper presents a well-structured benchmark study of encoder-decoder architectures and fine-tuning strategies for laparoscopic organ segmentation, supported by comprehensive ablation studies and practical insights.

It received two weak accepts and one weak reject. After reviewing the manuscript, reviewer comments, and author revisions, I recommend acceptance.

The authors are encouraged to further clarify the novelty/significance, discuss generalizability, and expand analysis of up-to-date transformer-based models.